# Modulatory Effects of Biosynthesized Gold Nanoparticles Conjugated with Curcumin and Paclitaxel on Tumorigenesis and Metastatic Pathways—In Vitro and In Vivo Studies

**DOI:** 10.3390/ijms23042150

**Published:** 2022-02-15

**Authors:** Satish Kumar Vemuri, Satyajit Halder, Rajkiran Reddy Banala, Hari Krishnreddy Rachamalla, Vijaya Madhuri Devraj, Chandra Shekar Mallarpu, Uttam Kumar Neerudu, Ravikiran Bodlapati, Sudip Mukherjee, Subbaiah Goli Peda Venkata, Gurava Reddy Annapareddy Venkata, Malarvilli Thakkumalai, Kuladip Jana

**Affiliations:** 1Sunshine Medical Academy Research and Technoloy (SMART), Sunshine Hospitals, PG Road, Secunderabad 500003, Telangana, India; banala.neuroscience@gmail.com (R.R.B.); kbvijayamadhuri@gmail.com (V.M.D.); drgpvsubbaiahgoli@gmail.com (S.G.P.V.); guravareddy@gmail.com (G.R.A.V.); 2Department of Biochemistry, Bharathidasan University Constituent College for Women, Tiruchirappalli 620009, Tamil Nadu, India; malarsai96@gmail.com; 3Division of Molecular Medicine, Centenary Campus, Bose Institute, P-1/12 C.I.T. Scheme VII-M, Kolkata 700054, West Bengal, India; satyajithalder1.bira@gmail.com; 4Biomaterials Group, Indian Institute of Chemical Technology (IICT), Tarnaka, Hyderabad 500007, Telangana, India; rahakrireddy@gmail.com; 5Global Hospital, Lakdi-ka-Pool, Hyderabad 500004, Telangana, India; m.chandrashekar005@gmail.com; 6Department of Biochemistry, Osmania University, Hyderabad 500007, Telangana, India; jeevanuttam@gmail.com; 7TBRC, Business Research Private Limited, Hyderabad 500033, Telangana, India; ravikiran.b03@gmail.com; 8Department of Bioengineering, Rice University, Houston, TX 77030, USA; sudip.mukherjee1988@gmail.com

**Keywords:** Curcumin, Paclitaxel, gold nanoparticles, triple-negative breast cancer cell lines (4T1 and MDA MB 231), metastasis

## Abstract

Background: Breast cancer is the most common cancer in women globally, and diagnosing it early and finding potential drug candidates against multi-drug resistant metastatic breast cancers provide the possibilities of better treatment and extending life. Methods: The current study aimed to evaluate the synergistic anti-metastatic activity of Curcumin (Cur) and Paclitaxel (Pacli) individually, the combination of Curcumin–Paclitaxel (CP), and also in conjugation with gold nanoparticles (AuNP–Curcumin (Au-C), AuNP–Paclitaxel (Au-P), and AuNP–Curcumin–Paclitaxel (Au-CP)) in various in vitro and in vivo models. Results: The results from combination treatments of CP and Au-CP demonstrated excellent synergistic cytotoxic effects in triple-negative breast cancer cell lines (MDA MB 231 and 4T1) in in vitro and in vivo mouse models. Detailed mechanistic studies were performed that reveal that the anti-cancer effects were associated with the downregulation of the expression of VEGF, CYCLIN-D1, and STAT-3 genes and upregulation of the apoptotic Caspase-9 gene. The group of mice that received CP combination therapy (with and without gold nanoparticles) showed a significant reduction in the size of tumor when compared to the Pacli alone treatment and control groups. Conclusions: Together, the results suggest that the delivery of gold conjugated Au-CP formulations may help in modulating the outcomes of chemotherapy. The present study is well supported with observations from cell-based assays, molecular and histopathological analyses.

## 1. Introduction

According to the published report by the International Agency for Research Cancer (IARC) on the global burden and the incidence of various cancer cases (36 types of cancers) in the year 2020 using a statistical database (i.e., Globocan 2020), around 19.3 million active cases and 10 million cancer-related deaths were reported in over 185 countries. Among these 19.3 million cases, the incidence of breast cancer cases was higher in comparison to other cancer cases [1].

Triple-negative breast cancers (TNBC) are a type of breast cancer characterized by a lack of cell receptors such as ER (estrogen receptor), PR (progesterone receptor), and human epidermal growth factor receptor 2 (HER2). TNBC is considered a dreadful and highly malignant cancer, due to the high chances of re-emergence and frequency at which it metastasizes in comparison to the other breast cancers. Currently, there are no effective therapies available against TNBC due to the lack of molecular targets in cancer cells, which underscores the need for developing novel therapies for this subtype of breast cancer [2]. The literature suggests the role of a multitude of proteins in tumor growth, cell–cell interaction, and metastasis. E-cadherin (E-cad) is such a protein whose suppression is linked with increased invasiveness and metastasis of tumors. TNBC molecular phenotype presents a higher risk for the loss of E-cad expression in comparison to the tumors of non-TNBC molecular phenotypes [3]. Metastasis is an impenetrable event that involves multiple interrelated biochemical events that makes it less lucid. Metastasis is divided into four essential steps: invasion, adhesion, detachment, and migration of different signaling pathways, and extracellular matrix regulated in cancer metastasis. The metastatic genes are stress-related genes that contribute to wound healing, inflammation, and stress-induced angiogenesis [4]. Yang et al. (2019) [5] demonstrated the crucial role of a signaling molecule known as signal transducer and activator of transcription 3 (STAT3) in evading the immune system, proliferation, and metastasis of TNBC5. TNBC continuously expresses the STAT gene for downregulating certain genes for gaining cellular responses by angiogenesis, anti-apoptosis, chemoresistance, immunosuppression, invasion, and migration. TNBC can be characterized by enhanced expression of vascular endothelial growth factor (VEGF) and vasculature [6]. Among the known chemotherapy drugs, Paclitaxel (Taxol) is the most extensively used chemotherapeutic drug for treating various cancers (i.e., breast, lung, ovarian cancer, and solid tumors). Paclitaxel exerts its anti-cancer effects by attaching itself to microtubules and inhibits cell division leading to induced apoptosis in cancer cells. However, the beneficial property of Paclitaxel as a chemotherapy drug has its limitations as it also causes adverse side effects including myelotoxicity and neurotoxicity [7] retinal vein occlusion, deep vein thrombosis, pulmonary embolism, stroke [8], and cataracts in cancer patients. Hence, the search for an alternate anti-cancer drug, which comes with the fewest side effects, is still on. Curcumin is a polyphenol known for its anti-inflammatory, anti-microbial, antioxidant, and anti-cancer properties [9,10]. Curcumin is known to act against many signaling molecules, including growth factors (VEGF-a/b/c/d), transcription factors, kinases, metalloproteases (MMP1/2) cytokines, and other receptors associated with angiogenesis and angiogenesis-dependent metastasis of tumor cells by downregulating respective genes [11]. Therefore, based on the extensive research findings regarding phytochemicals, and their derivatives, this can be a promising option for cancer treatment with lower toxicity and effectiveness [12,13]. After the selection of the phytochemicals, the focus shifts to the delivery systems that are inert and apt in delivering the drugs to the site of cancers. Scientists have researched varied delivery systems such as nanoparticles (silver and gold), liposomes, etc., both in vitro and in vivo for utilizing better therapeutic efficacies against different cancers, metastatic and drug-resistant tumors. Among the nanoparticles, gold nanoparticles (AuNPs) were extensively used as drug delivery systems (DDS) due to their easy synthesis, high surface volume, easy functionalization, and high biocompatibility. Nanoparticles exhibit better efficacy and a censorious role in drug delivery [3,14]. Our earlier publication on the evaluation of biosynthesized gold nanoparticles (i.e., Au-Cur, Au-Tur, Au-Qu, and Au-Pacli) alone and in combination in vitro has generated encouraging results giving us evidence on the synergistic effect of phytochemicals and Paclitaxel in employing significant anti-cancer properties with the regulation of gene expression playing a vital role in disease progression [15,16]. Based on the evidence from our earlier studies, we developed an in vivo breast cancer model (i.e., Balb/c mice) by injecting the 4T1 cell line into the fourth inguinal (lower mammary fat pad) [17,18,19,20] of the mice for evaluating the anti-cancer properties of experimental drugs (Curcumin and Paclitaxel) and biosynthesized gold nanoparticles (Au-C, Au-P, and Au-CP) individually and in combination. The current study evaluates the anti-metastatic and synergistic effects of Curcumin and Au-CP in metastatic triple-negative breast cancer cell lines and mouse breast cancer models. Findings from the study exhibited anti-cancer and anti-metastatic properties of b-AuNPs (i.e., Curcumin alone and in combination with Paclitaxel) by controlling the expression of pro-oncogenes (MMPs, STAT, VEGF, Cyclin D) and enhancing the expression of the apoptotic gene (Caspase 9), anti-metastatic gene (E-cadherin), reduction in migration of cells in vitro, and reduction in the tumor size (in vivo). Based on the findings, it is evident and safe to promote to the phytochemical combinatorial chemotherapy approach in treating the triple-negative breast cancers as could be the future of nanomedicine in the field of cancer treatment.

## 2. Results

### 2.1. Characterization of Gold Nanoparticles (AuNPs) and Stability Study

After synthesis of drug conjugated gold nanoparticles, initially, we evaluated physicochemical parameters such as hydrodynamic diameter (HDD), zeta potentials, and polydispersity index (PDI) of Au-C, Au-P, and Au-CP by using dynamic light scattering spectroscopy (DLS) in various media (Mili-Q water and 10% serum containing media has a lot of BSA (bovine serum albumin) proteins, these proteins may bind to the surface of the gold nanoparticles and it may help to increase the surface diameter of the gold nanoparticles [21,22] (Table 1).

We observed that these particles were in the nanometer range in water (101–128 nm) (Figure 1a, middle panel and Table 1) and serum-containing medium (Table 1) and negative zeta potentials in both the mediums (Figure 1a, lower panel and Table 1). Transmission electron microscopic images clearly showed that all three formulations were circular and uniformly distributed (Figure 1a, upper panel).

The stability of the particles was analyzed in water for up to 100 days. We observed that there was no significant change in the particle HDD (Figure 1b) indicating high stability of the nanoconjugates that are essential for any biomedical applications.

### 2.2. Apoptotic Assay

The observations from the apoptotic assay highlight the treatments with Cur, Pacli, Au-C, Au-P, CP, and Au-CP, individual 10.0 μg/mL and in the combination of AuNPs-CP at 5.0 μg/mL for 36 h. The triple-negative metastatic cancer cell lines (4T1 and MDA MB 231 cell lines undergoing apoptosis) were quantified by the Annexin-V binding assay. The Annexin-V (apoptotic cells) phosphatidylserine (dead cells) showed 4T1 and MDA MB 231 cancer cells enhanced the apoptosis on treatment with the experimental drugs and the percentages of dead cells in the MDA MB 231 cell line are as follows: Cur (34.03% early apoptosis cell and 0.70% late apoptosis cell), Pacli (16.18% early apoptosis cell, 6.92% late apoptosis cell, and 4.96% necroptosis cell population), CP (52.40% early apoptosis cell and 12.67% late apoptosis cell), Au-C (32.57% early apoptosis cell, 18.22% late apoptosis cell, and 2.18% necroptosis cell population), Au-P (10.29% early apoptosis cell, 23.34% late apoptosis cell, and 5.74% necroptosis cell population), and Au-CP (38.36% early apoptosis cell, 35.20% late apoptosis cell, and 1.58 necroptosis cell population).The percentage of dead cells in the 4T1 cell line are as follows: Cur (26.39% early apoptosis cell, 4.67% late apoptosis cell, and 0.76% necroptosis), Pacli (17.87% early apoptosis cell, 6.54% late apoptosis cell, and 1.10% necroptosis cell population), CP (29.48% early apoptosis cell, 20.60% late apoptosis cell, and 6.32% necroptosis cell), Au-C (39.38% early apoptosis cell, 4.77% late apoptosis cell, and 0.58% necroptosis cell population), Au-P (32.36% early apoptosis cell, 2.20% late apoptosis cell, and 1.63% necroptosis cell population), and Au-CP (43.52% early apoptosis cell, 28.76% late apoptosis cell, and 0.09 necroptosis cell population). The results showed significant cytotoxicity in both 4T1 and MDA MB 231 cell lines on treatment with individual agents and also in combination, but the combination treatments demonstrated better apoptotic efficiencies in comparison to the individual treatments (Figure 2a,b) [23,24,25,26,27].

### 2.3. DCFDA Assay

To determine whether Cur, Pacli, Au-C, Au-P, CP, and Au-CP-mediated apoptotic cell death was due to intracellular reactive oxygen species (ROS) accumulation, we used an H2DCFDA fluorescence assay and ROS generation was measured using fluorescence microscopic analysis. Fluorescence microscopic images of treated MDA MB 231 and 4T1 cells showed enhancement in the level of green color fluorescent intensity (Cur (6.5 and 6.5), Au-C (10 and 10), Pacli (4.5 and 4.5), Au-P (8 and 7), CP (18 and 16), and Au-CP (22 and 18)) as compared to control untreated cells indicating intracellular ROS accumulation. The combination therapies demonstrated significant intracellular ROS accumulation in both 4T1 and MDA MB 231 cell lines in comparison to the individual treatments (Figure 2c,d) [28,29].

### 2.4. Retardation in MDA-MB 231 Cell Migration

To further investigate whether anti-proliferative changes in Cur, Pacli, Au-C, Au-P, CP, and Au-CP treated cells affected the migratory activity of MDA-MB 231 and 4T1 cells, a bidirectional scratch assay was performed. Decreased cellular migration in the wound areas which were treated with Cur, CP, Au-C, and Au-CP was observed, indicating that they could inhibit the mobility of breast cancer cells. The cells treated with Pacli and Au-P showed the least effect as a result of lesser inhibition of cellular migration in the wound area in comparison to control cells that gradually migrated and filled the wounded area after 24 h of treatment. To confirm our above-mentioned findings, the transwell migration assay was performed using Cur and Pacli (10.0 μg/mL) individually and in the combination of Au-CP at 5.0 μg/mL for 36 h. Consistent with our findings from the scratch assay, data from the transwell migration assay also showed in response to Cur, CP, Au-C, and Au-CP, lower numbers of cells migrating to the undersurface transwell insert in comparison to a significantly high number of migratory cells that migrated to the undersurface transwell insert in Pacli and Au-P-treated and control sets (number of cells migrated in different treatment groups, i.e., control (2500 and 1800), Cur (1250 and 1000), Au-C (600 and 600), Pacli (1600 and 1000), Au-P (800 and 700), CP (300 and 450), and Au-CP (200 and 300)). Thus, these observations indicate that Curcumin in combination could be a very effective treatment regimen for controlling breast cancers (Figure 3a,b) [30].

### 2.5. Flow Cytometer Analysis—E-Cadherin Expression in 4T1 and MDA-MB 231 Cell Lines

The below figure illustrates the efficacy of the anti-cancer treatments for controlling metastasis by enhancing the expression of E-cadherin in MDA-MB 231 and 4T1 cancer cells. E-cadherin expression signal was more evident in Cur, Pacli (10.0 μg/mL), and the combination of Au-CP at 5.0 μg/mL for 36 h treatments in comparison to the control group. The expression signals of E-cad in Pacli and Au-P treatment groups (MDA MB 231 and 4T1) were low in comparison to the other treatments groups (i.e., Cur, Au-C, CP, Au-CP) suggesting low efficacy against metastasis (Figure 3c,d) [31,32].

### 2.6. Scratch Assay

To determine the efficacy of different anti-cancer agents against cancer cell motility in vitro, the scratch assay was employed at different time points (0 and 24 h). The wound that was created was considered time zero. MDA-MB-231 cell migration was observed for 24 h and it was observed that the wound area increased in the cells, which were treated with Cur, CP, Au-C, and Au-CP, indicating that they could inhibit the motility of breast cancer cells. The cells treated with Pacli and Au-P showed the least effect as a result as the wound area appeared smaller which was similar to the control group [5,33]. Cur and Pacli at 10.0 μg/mL and the combination of Au-CP at 5.0 μg/mL for 36 h inhibited cell migration in MDA-MB-231 indicating that Curcumin in combination could be a very effective treatment regimen for controlling breast cancers (Figure 4a,b) [34].

### 2.7. Gene Marker Studies

MDA-MB 231 and 4T1 cell lines were treated with Cur, Pacli, CP, Au-C, Au-P, and Au-CP to assess their effect on the expression of Caspase 9, VEGF, STAT, and Cyclin D gene markers. The expression of different gene markers in the cells were treated individually and in combination with Cur, Pacli, CP. We observed that the combinational treatments significantly downregulated VEGF, CYCLIN D1, and STAT-3 genes and upregulated the Caspase-9 expression in comparison to the individual treatments in both 4T1 and MDAMB 231 cell lines. The individual and combinational treatments gave significant results in the case of the 4T1 cell line when compared to the MDA MB 231 cell line (Figure 4c,d) [30].

### 2.8. Tumor Induction and Treatment

A reduction in tumor size (primary xenografts) in BALB/c mice was observed. Curcumin decreased tumor size in primary breast cancer xenografts individually and in combination with Paclitaxel. BALB/c mice bearing 4T1 cells as xenografts were intraperitoneally treated with 50 mg/kg Curcumin and Paclitaxel daily for 3 weeks. Tumors in Cur, Pacli, and CP-treated mice were 50% of the size of control animals at the end of drug treatment. Xenografts were introduced into the breast fat pads of Balb/c mice and in a week the tumors were visible. The treatment with the Curcumin and Paclitaxel alone and in combination showed a reduction in tumor size. The mice that received Curcumin and CP combination showed a significant tumor reduction in comparison to the control and Pacli-treated mice. The use of AuNPs increased the therapeutic efficacy in all groups possibly due to the higher availability of drugs at disease sites using enhanced permeability and retention phenomena (Figure 5a) [35,36,37]. The size of tumors was measured externally in the experimental mice using a Vernier caliper at different time points (days 0, 10, and 21) and it was observed that the tumors regressed with the experimental treatments. The tumors significantly regressed in mice that received Cur, CP, Au-C, Au-CP, indicating the effectiveness of the treatments against breast cancers individually with Cur and Au-C and also in combination CP and Au-CP with the known anti-cancer drug Paclitaxel. The mice that received Paclitaxel showed minimal effects on tumors in comparison to other treatments, but the effect increased significantly when used as a combination with Curcumin, demonstrating the synergistic and beneficial effects in controlling tumor progression and disease (Figure 5b).

### 2.9. Histopathological Studies

Figure 5c demonstrates the histopathological evaluation of the breast and hepatic tissues of experimental mice for the status of metastasis and the potency of experimental drugs for controlling the progression of cancer and metastasis in vivo. Metastasis was prominently seen in the positive control, Pacli, and Au-P groups. Mild metastasis was observed in the groups that received Cur and Au-C, but the groups that received a combination of CP and Au-CP showed no signs of metastasis (i.e., neoplasticity in both mammary and hepatic tissues) indicating the effective therapeutic potential of CP and Au-CP [35].

## 3. Discussion

Over the years, focus has shifted towards phytochemical supplementation in cancer management that has become acceptable and cost-effective, but there is limited evidence that plant-derived constituents could decrease the risk or prognosis of cancer. In the last two decades, varied non-nutritive phytochemicals were isolated individually or as a mixture of agents, and their potency as chemo-preventive agents was evaluated. Despite scientific advancements in understanding of the incidence and progression of cancer, limitations in elucidating the chemo-preventive abilities of most phytochemicals are still encouraging researchers to conduct extensive research in this area of therapy. Anti-tumor activities of most phytoextracts are due to the combination of varied phytochemicals acting synergistically against carcinogenesis rather than an individual agent. It is hypothesized that an individual agent may not exert its anti-tumor properties similarly or may lose its bioactivity when isolated from the whole compound [36,37]. The challenge for researchers is the identification and characterization of compounds, molecular and cell signaling networks for better assessment of phytochemicals and the dosage to be administered to humans in a day.

The combination of Curcumin–Paclitaxel is potent for controlling the cancers by inhibiting the cell proliferation, metastasis, and enhancing the pro-apoptotic markers such as P53 and caspases 3, 7, 8, and 9 [38].

Gold nanoparticles are behaving as a delivery system as they can overcome the biological barriers, sustain the blood flow for a long duration, reach target specific cancer cells in distant areas, and accumulate densely in the tumor sites and release drugs [39].

The present study demonstrated the anti-cancer and anti-metastasis mechanisms of Curcumin, Paclitaxel, gold nanoparticle conjugated Curcumin, and Paclitaxel on metastatic cancers individually and in combination. Our earlier study reported the anti-metastatic properties of nanoparticle conjugated phytochemicals by targeting the STAT3, Cyclin D, and VEGF pathways in vitro [40,41]. STAT3 and MMPs hold a key role in proliferation and evasion of the apoptotic pathway in metastatic breast cancers. The authors of [42] reported on the tumor cells undergoing apoptosis, blebbing, necrosis, nuclear fragmentation, and formation of apoptotic bodies upon treatment with pectolinarigenin. Bcl-2 proteins are known as apoptosis regulators as they regulate the caspase-9 and caspase-3 genes in an apoptotic cascade and caspase-2/9 were mostly found in apoptotic pathways [43]. Observations in this study revealed that Cur, Pacli, CP, Au-C, Au-P, and Au-CP inhibited the metastasis of negative breast cancer cells by downregulating STAT3, MMP2/9, and cyclin D-1 and induced apoptosis in both in vitro and in vivo models and our findings co-relate with the literature [40,41,42,43].

TEM analysis was carried out to determine the size of monodispersed spherical nanoparticles of the Au-C and Au-P. The average size of Au-C and Au-P was found to be 5–40 nm (Figure 1a). DLS studies were carried out to calculate the hydrodynamic diameter and zeta potential. DLS results showed that the sizes of the nanoparticles for Au-C and Au-P were 101 and 128 nm; zeta potential was −0.2 ± 0.2 mV for Curcumin and 15.5 ± 0.9 mV for Paclitaxel (Figure 1b). It is important to mention that the size of the b-AuNPs obtained from DLS is greater than the size obtained from TEM [15].

The hydrodynamic diameter of AuNPs is shown to be higher in DLS in comparison to TEM analysis because DLS considers the water surrounding the nanoparticles and also the coated phytochemicals, but TEM considers only the metallic part of the nanoparticles. We also measured the zeta potential of Au-C and Au-P, which provides the surface charge or surface potential of the nanoparticles. High positive or negative zeta potential data indicates the colloidal stability of nanoparticles. The result showed negative zeta potentials (Au-C: −0.2 ± 0.2 mV and Au-P: −5.8 mV ± −6.7 mV) for Au-C and Au-P that supports the colloidal stability of the nanoparticles (Table 1 and Figure 1b).

An MTT assay was performed to demonstrate the cytotoxic properties of Cur, Pacli, CP, Au-C, Au-P, and Au-CP in breast cancer cell lines (i.e., MDA MB 231, 4T1, and HEK 293). Cytotoxicity was successfully induced by phytochemical combination of anticancer drugs at 10–20 μg/mL Cur, Pacli, and combinations of CP and Au-CP (5.0–10 μg/mL) concentrations in breast cancer cell lines. It was observed that combination treatments exerted better cytotoxicity in both MDA-MB 231 and 4T1 cell lines when a comparison was made between these individual and combination treatments in cell lines. Similarly, when these treatments were examined on the normal cell line (HEK-293) they did not exert any cytotoxic effects. Hence, we conclude that Cur, CP, Au-C, and Au-CP (10–30μg/mL) selectively target the cancer cells. Our results are in agreement with earlier findings (Figure 1c).

Gene marker studies (qPCR) revealed the anti-metastatic and apoptotic properties Cur, Au-C, CP, Au-CP in both triple-negative cell lines (4T1 and MDA MB 231) by under-expressing the VEGF, Cyclin D, and STAT3 genes and enhancing caspase 9 gene expression. However, Pacli and Au-P treatments showed minimal effects on the expression profile of metastatic genes when compared to other treatment groups and their expression profile was similar to that of controls. Based on the results, it is convenient to say that Curcumin alone and in combination effectively controlled the metastatic process in comparison to the Paclitaxel treatment alone. Similar findings were observed in our in vivo studies (qPCR) which supported our in vitro findings. Based on the results, it is convenient to say that Curcumin alone and in combination effectively controlled the metastatic process in comparison to the Paclitaxel treatment alone. Our data is in agreement with existing literature [44].

Similar studies were reported, illustrating the role of nanoparticles in the treatment of various cancers by inducing increased apoptosis signals and activation of ER stress and Ca^2+^ signaling pathways. Varlamova et al. (2021) and Turovsky and Varlamova (2021) reported on cytotoxicity effects of Selinium nanoparticles (SeNPs) in cancer cells (A-172 and MCF-7) by activating the mitochondrial apoptosis pathway and by overexpression of pro-apoptotic gene markers such as BCL-2 and caspase 3 and by releasing cytochrome-C into the cytoplasm of cells by increasing the permeability of the mitochondrial membrane, when treated with 5 µg/mL SeNP. As a consequence of SeNP treatment, higher expression of the caspase 4 gene in A-172 cells indicates modulation of the intrinsic ER-mediated pathway of apoptosis and Ca^2+^ signaling pathways [45,46].

Flow cytometric analysis of E-cadherin and apoptosis in MDA-MB-231 and 4T1 metastatic breast cancer cell line was undertaken after treatment with Cur, Pacli, CP, Au-C, Au-P, and Au-CP, and it was observed that combination treatments were exerting better cell apoptosis and inhibiting the metastatic E-cadherin pathway in both breast cancer cell lines in comparison with either substance alone in cancer cell lines. Therefore, breast cancer treatment may benefit from the use of a combination of drugs in chemotherapy (Figure 3c,d).

Observations from the in vitro study suggest the synergistic efficacy of Curcumin in combination with Paclitaxel for controlling the TNBC cells from metastasizing. Hence, the combination therapy could be a potential treatment regimen for controlling metastatic breast cancers (Figure 4c). Numerous in vitro and vivo studies reported on the combination of Paclitaxel with phytochemicals and nanoparticles exhibiting synergistic anticancer effects in the treatment of breast cancer [47,48,49]. The physical and topical examination of induced tumor in the breast tissue revealed significant progression of disease and size of the tumor in the positive control, Pacli and Au-P groups, whereas reduced tumor size was observed in groups that received individual treatments with Cur and Au-C, but the groups that received combinational treatments (CP, Au-CP) demonstrated the efficacy of combinational therapy against the induced tumor progression and metastasizing tumor cells (Figure 5a,b).

The status of metastasis in the breast and hepatic tissues of experimental mice was assessed. The neoplastic cells (Grade 3) with variation in size, shape, and multiple mitotic figures were also observed in the brain and liver of positive control and Paclitaxel-treated groups (Pacli and Au-P) indicated by arrows [50,51,52,53,54,55,56,57,58]. The mice that received Cur and Au-C showed the fewest neoplastic cells in mammary glands, whereas the observation in liver tissue revealed the presence of multiple nodular tumor mass with poorly differentiated neoplastic cells which were indicated by arrows. Whereas, the mice treated with CP and Au-CP showed normal cell morphology in the subcutaneous region in the mammary gland and normal portal triad with the portal vein and bile duct in the liver tissues (indicated by an arrow) (Figure 5c).

Our earlier studies demonstrated the apoptotic, anti-angiogenesis, anti-proliferative, anti-colony formation, and anti-spheroid formation properties of biosynthesized gold nanoparticles (i.e., Au-Cur, Au-Tur, Au-Qu, and Au-Pacli) when treated alone and in combination with Paclitaxel in vitro and provided evidence of the synergistic effects of phytochemicals and Paclitaxel in exerting significant anti-cancer effects by controlling the expression of various genes playing roles in disease progression [15]. The presence of Curcumin (non-conjugated and conjugated with gold nanoparticles) will enhance the anti-cancer properties of Paclitaxel even in the drug resistant and TNBC by activating different apoptotic pathways and reducing the expression of different metastatic genes. The biosynthesis of gold nanoparticles is an easy and economical process in reducing aurum chloride by using phytochemicals in the presence of heat. Developing an in vivo model for evaluating the potential therapeutic agents is very crucial for understanding the anti-metastatic mechanism in real-time. Therefore, it is speculated that the combination of Paclitaxel and Curcumin with gold nanoparticles may be an ideal strategy in clinical practice for cancer treatment. The advantages include lower cost, faster, and minimum ethical concerns.

## 4. Materials and Methods

### 4.1. Chemicals and Reagents

Chemicals and reagents were obtained from Merck (Mumbai, India), Himedia Mumbai, India), Invitrogen (Benguluru, India), SRL (Hyderabad, India), DCFDA (#D6883) was purchased from Sigma-Aldrich (Benguluru, India). Fetal bovine serum (#16000044) was obtained from Gibco, Cleveland, TN, USA, and MEM sodium pyruvate, MEM non-essential amino acids L-glutamine and Gentamicin, were procured from Hi-Media, India.

### 4.2. Preparation of Stock Solutions

A stock solution of 10^−2^ M was prepared by dissolving 1 gm of HAuCl_4.3_H_2_O in 253.92 mL of autoclaved Milli-Q water (Millipore). We further prepared stocks of 10 mg/mL of Curcumin in 1 mL of DMSO. Moreover, we prepared a stock of Paclitaxel at a concentration of 10 mg/mL.

### 4.3. Synthesis of Gold Nanoparticles Using Phytochemicals (Curcumin and Paclitaxel)

HAuCl4 (200 μL, 10^−2^ M) solution was added to a 4.8 mL water fraction of Curcumin and Paclitaxel (10 μg/mL). The total volume of the reaction mixture was adjusted to 5 mL in all experiments. The resulting Au-C and Au-P were purified by ultracentrifugation at 15,000 rpm (25,000× *g*) for 40 min at 15 °C (Sorvall WX ultra-100, Thermo scientific, Benguluru, India) [15,18].

#### Physicochemical Characterization of Curcumin and Paclitaxel Conjugated Gold Nanoparticles

Hydrodynamic diameter (HDD), zeta potentials, and poly-dispersity index (PDI) of Au-C, Au-P, and Au-CP were analyzed by photon correlation spectroscopy using a Lite Sizer TM 500 Particle Analyzer, manufactured by Anton Paar. The HDD, charges, and PDI of the particles were analyzed in deionized water and a 10% serum-containing medium. The stability of the particles was measured in deionized water up to 100 days DLS. The size and morphology of the particles were examined using transmission electron microscopy (TEM), Tecnai G2 F30 S-Twin Microscope, operated at 100 kV. Selected area electron diffraction patterns were also recorded using this instrument. Inductively, each experiment was carried out in triplicate in three different batches [15,18].

### 4.4. Cell Culture

The 4T1 cell line, mouse epithelial triple-negative breast metastatic cell line and MDA-MB 231 cell line, a human epithelial breast cancer cell line (4T1 cells and MDA-MB 231) were aseptically cultured in multi-well cell culture plates using Dulbecco’s Modified Eagle’s Medium, (DMEM High Glucose) supplemented with 10% fetal bovine serum, 50 units/mL penicillin, and 50 μg/mL streptomycin. Then, 24 h after seeding, these cell lines were treated with 10.0 μg/mL of Cur, Pacli, Au-C, and Au-P (optimized dose) when treated alone, and 5.0 μg/mL of CP and Au-CP as combination therapy. The 4T1 and MDA-MB 231 cell lines were processed for various tests at indicated post-treatment periods (36 h for cell apoptosis; 36 h for quantitative PCR) [15,19].

### 4.5. Cell Viability Assay/MTT Assay

4T1, MDA-MB 231, and HEK293 cell lines were plated in 96-well plates for cell viability assay. The cell lines were treated with vehicle control or different doses of Cur, Pacli, Au-C, and Au-P (10 and 20 μg/mL), and with combinations of CP and Au-CP (5 and 10 μg/mL) for 36 h. A dose of 5 μg/mL was selected for combination therapy which is precisely half of the dose of the pristine nanoconjugates. The reason behind this selection is to avoid unwanted cytotoxicity in normal cells as in the combination of b-AuNPs it is much more cytotoxic than its pristine form. After 36 h, cell viability was analyzed using the MTT assay using a published protocol and absorbance was recorded at 475 nm concerning 660 nm [20].

### 4.6. Annexin V-FITC/PI Staining for Apoptosis Assay

Induction of apoptosis was quantified via flow cytometric analysis of control and Cur, Pacli, CP, Au-C, Au-P, and Au-CP-treated cells that were stained with Annexin V-FITC/PI using the Annexin V-FITC apoptosis detection kit according to the manufacturer’s protocol (BD Bioscience, San Jose, CA, USA). Briefly, 4T1 and MDA-MB-231 cells treated with compounds for 36 h were subjected to Annexin-V assay to quantify the number of apoptotic cells by flow cytometry. Cells were further trypsinized, washed once in PBS, and the pellets were collected after centrifugation and resuspended in binding buffer and to this 5 μL fluorescein isothiocyanate (FITC)-labeled Annexin-V and 10 μL propidium iodide (PI) were added. The resulting mixtures were incubated in the dark for 10 min at room temperature and the fluorescence of the cells was determined immediately using BD FACS Verse flow cytometer (BD Biosciences, San Jose, CA, USA). Annexin V/FITC positive cells were regarded as apoptotic cells analyzed using Cell Quest Software (BD Biosciences) [19].

### 4.7. Measurement of Cellular ROS Using DCFDA

To estimate the intracellular reactive oxygen species (ROS) production due to Cur, Pacli, CP, Au-C, Au-P, and Au-CP treatment, the DCFDA method was used (2, 7-Dichlorodihydrofluorescein-diacetate). Briefly, MDA-MB 231 and 4T1 cells were seeded in a 6-well plate and treated with test compounds (10 mg/mL) for different periods. Post-treatment, the media was discarded and incubated with 10 µM H2DCFDA for 30 min at 37 °C. For fluorescent imaging, H2DCFDA incubated cells were washed, resuspended in 1x PBS, and directly imaged under a fluorescent microscope (Leica) [18,19].

### 4.8. Transwell Migration Assay

To assess cell migration and invasion capacities of Cur, Pacli, CP, Au-C, Au-P, and Au-CP-treated MDA-MB 231 and 4T1 cells, a transwell assay was undertaken using cell culture inserts with a pore size of 8 µm. Briefly, cell culture inserts were placed in 12-well companion plates containing 300 µL of serum-free DMEM media and MDA-MB 231 and 4T1 cells at a density of 2 × 10^5^ cells were seeded in the upper half of the insert. Then, 700 µL DMEM media containing 10% FBS were then added into the lower chamber and treated with a respective concentration of compounds for 24 h. Post-treatment, the inserts were removed and the cells on the upper surface of the membrane were wiped off with cotton swabs. Then, the cells that had invaded into the micro-porous membrane were washed three times with 1X PBS, fixed with 3.7% formaldehyde, and were permeabilized using methanol. At last, cells were stained with Giemsa stain for 30 min, and the cells were observed with a microscope, and images were obtained. Cells present in the lower part of the inserts were counted in three microscopic fields per well, and the extent of migration was expressed as an average number of cells per microscopic field [19].

### 4.9. E-Cadherin Expression by Flow Cytometry

MDA-MB 231 and 4T1 cell lines were harvested after trypsinization then washed with 1X PBS and stained with primary antibody for E-cadherin, incubated for 30 min on ice at room temperature. After incubation, cells were then washed three times. E-cadherin and the control samples were then incubated with anti-rabbit-FITC IgG antibodies for 30 min under the same conditions. Cells were washed again, and flow cytometric analysis was performed [20].

### 4.10. Scratch Assay

A bidirectional scratch assay was used to determine the migration of breast cancer cells MDA-MB 231 and 4T1 under different conditions. Firstly, 4T1 and MDA-MB 231 cells were cultured in 6-well plates for 24 h to achieve 100% confluence followed by starvation in serum-free DMEM. After which, a scratch was done using a 200 µL sterile pipette tip to form a bidirectional wound. Cells were then treated with Cur, Pacli, CP, Au-C, Au-P, and Au-CP for different time periods (0 and 24 h). The migration width at each time point in each treatment group was measured at four different positions when microscopic images of the cells were captured and compared with the gap width at 0 h [19].

### 4.11. Quantitative RT-PCR

Total RNA was extracted with an RNASure mini-isolation kit Nucleo-pore, Genetix, and RNA (1 µg) was converted to cDNA using a Thermo fisher cDNA synthesis kit. The analysis of gene expression was performed using gene-specific primers (Table 1). The qRT-PCR steps were as follows: (1) denaturation at 95 °C for 3 min, (2) 30 cycles at 95 °C for 1 min, (3) 57 °C for 30 s (depending on primer sets), (4) 72 °C for 1 min, and (5) extension at 72 °C for 7 min. The melting curve examination verified a single product. Relative expression quantities were evaluated and normalized by comparing them to action [19,20].

### 4.12. Tumor Induction in Mice

Our study adopted the methodology and therapeutic strategies of [19] in developing the induced breast tumor model for safety and efficacy studies of our treatments. For the present study we used 8–10 weeks old female BALB/c mice, weighing about 25–32 g, for developing the tumor model. For developing the model, we sedated the mice by administering ketamine and xyalzine (as per animal weight and accepted norms). Once the animals were completely in sedation, we identified the mammary gland (i.e., 3rd and 4th glands in abdomen area) and injected a low count of 4T1 breast cancer cells (1 × 10^4^ cells) into the BALB/c mouse mammary fat pad using a tuberculin syringe. Weekly assessments of weight and tumor size measurements, food and water intake were completed.

### 4.13. Statistical Analysis

The results were expressed as mean ± SEM. One-way ANOVA was followed by Tukey’s comparison test. The level of significance was set at **** (*p*-value: <0.0001) *** (*p*-value: <0.001); ** (*p*-value: ≤0.01); * (*p*-value: 0.05) in respect to the control. GraphPad Prim 5.0 software was used to statistically analyze the data.

## 5. Conclusions

The current study demonstrated the inhibitory role of various individual and combinational treatments on different triple-negative metastatic breast cancers (in vitro and in vivo). The treatments controlled or reduced the progression, proliferation, and metastasis of cancer cell lines. Better efficacy was observed when drugs were delivered using gold nanoparticles that acted as a delivery vehicle and helped to provide targeted therapy. The present study was a comparative study between the individual and combination treatments to evaluate the synergistic activity of Curcumin and Paclitaxel for controlling the metastasis and disease progression. This study also provided rationale for considering the use of Curcumin, gold conjugated–Curcumin individually or in combination with Paclitaxel as an alternative therapeutic strategy to the existing chemotherapy regimens. The combination of Curcumin–Paclitaxel might help by ameliorating the side effects caused by Paclitaxel such as oxidative stress and induced cytotoxicity in cardiomyocytes. Further studies are still required before taking it into the clinics.

## Figures and Tables

**Figure 1 ijms-23-02150-f001:**
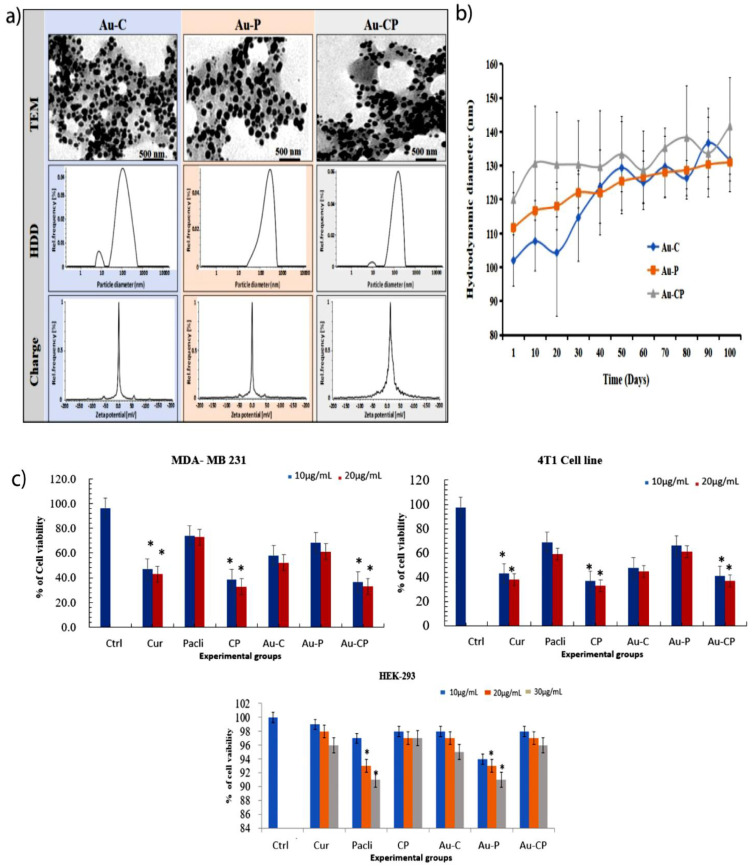
Cytotoxicity and cell viability assessment (MTT assay). (**a**) Characterization of gold nanoparticles conjugated with Cur, Pacli, and CP. Upper panel: Transmission electron microscopic (TEM) images of Au-C, Au-P, and Au-CP (scale bar 500 nm). Middle panel: Hydrodynamic dynamic diameter histograms of Au-C, Au-P, and Au-CP obtained by DLS in water. Lower panel: Zeta potential histograms of Au-C, Au-P, and Au-CP obtained by DLS in water. Each formulation was prepared and characterized thrice, and representative histograms are shown. (**b**) Stability analysis of Au-C (blue line), Au-P (orange line), and Au-CP (grey line) in deionized water at different time points at room temperature. (**c**) MTT assay (cytotoxicity and cell viability) was performed in cancer cell lines MDA-MB-231, 4T1 by treating with 10.0 and 20.0 μg/mL individually and in the combination of Au-CP at 5.0 and 10.0 μg/mL for 36 h. Similarly, HEK-293 cells were treated with three concentrations (10, 20, and 30 μg/mL). *p*-value < 0.05, * indicates significance.

**Figure 2 ijms-23-02150-f002:**
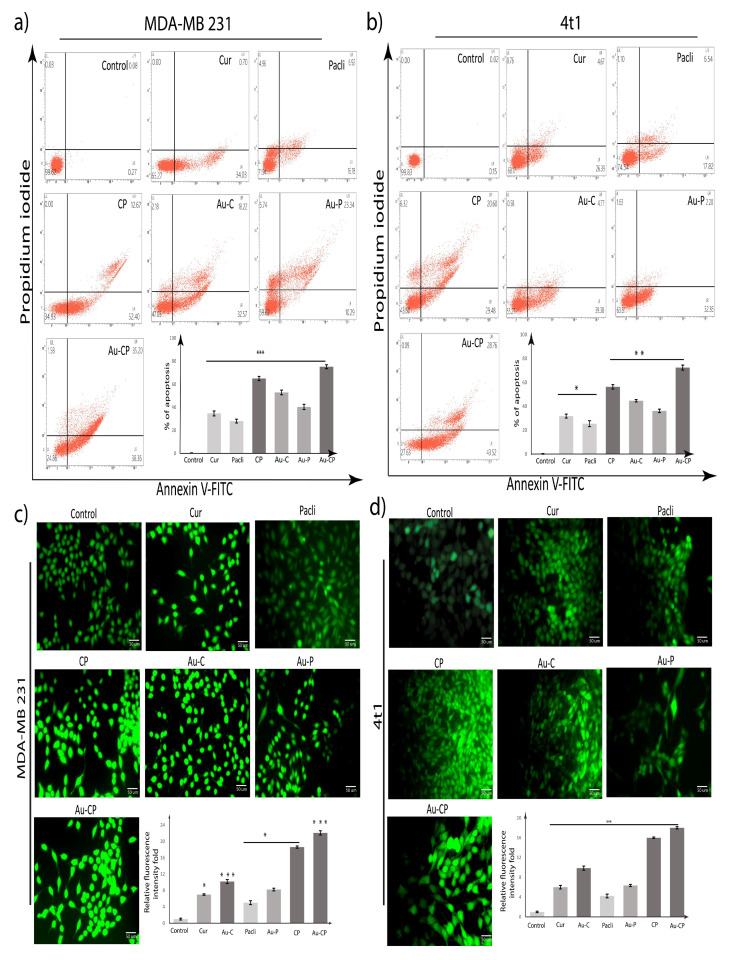
(**a**,**b**) Apoptotic assay—percentage of cells undergoing apoptosis in different breast cancer cell lines, (**a**) MDA MB 231 and (**b**) 4T1, after treatment with Cur, Pacli 10.0 μg/mL, and in the combination of Au-CP at 5.0 μg/mL for 36 h; Annexin V–PI assay). (**c**,**d**) DCFDA assay—fluorescence microscopic images of treated (**c**) MDA-MB 231 and (**d**) 4T1 cells showed enhancement in the level of green color fluorescent intensity as compared to control untreated cells indicating intracellular ROS accumulation. *p*-value: <0.05, 0.01, 0.001 indicates * significant, ** very significant, *** highly significant.

**Figure 3 ijms-23-02150-f003:**
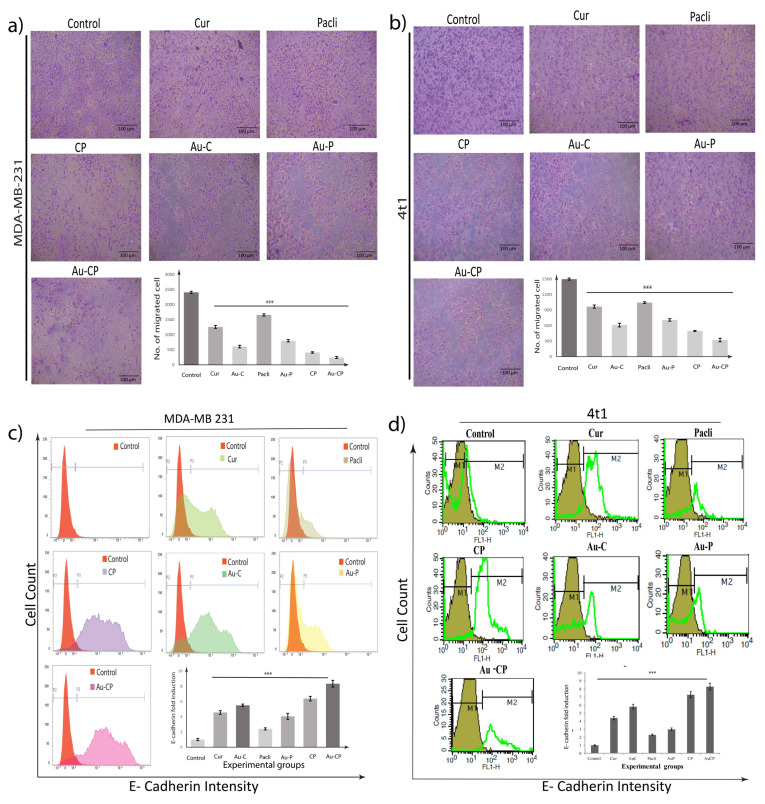
(**a**,**b**) Transwell migration assays of (**a**) MDA-MB 231 and (**b**) 4T1 were performed to assess the motility of metastatic cancer cells upon treatments with Cur and Pacli individually and both in combination CP in a transwell chamber with the non-coated membrane (24-well insert, pore size: 8 mm, Corning, Life Sciences). (**c**,**d**) Flow-cytometric analysis of E-cadherin in (**c**) MDA-MB 231 and (**d**) 4T1 metastatic breast cancer cell lines on treatment with Cur, Pacli at 10.0 μg/mL and the combination of AuNPs-CP at 5.0 μg/mL for 36 h. *p* < 0.001 *** signifies highly significant.

**Figure 4 ijms-23-02150-f004:**
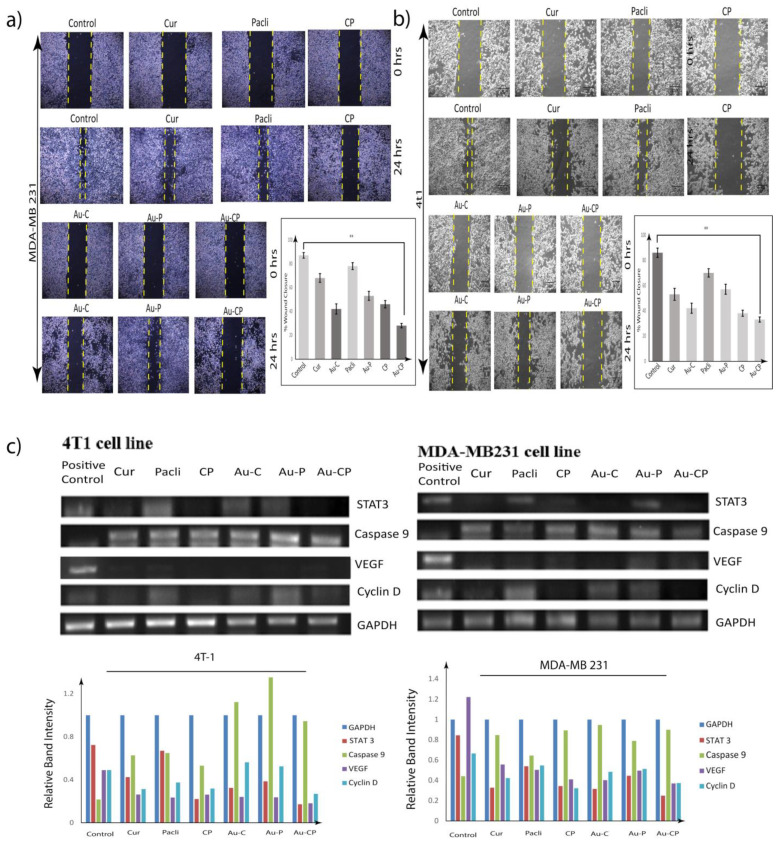
(**a**,**b**) Scratch assay—analysis of (**a**) MDA-MB 231 and (**b**) 4T1 cell migration by in vitro scratch assay upon treatment with sub-toxic concentrations of Cur and Pacli individually and both in combination CP, as indicated. Images were acquired under a phase-contrast microscope at 0 and 24 h. The dotted lines define the areas lacking cells. Cur, Au-C, CP, and Au-CP inhibited cell migration in MDA-MB-231 whereas Pacli and Au-P did not inhibit the cell migration. (**c**) The gene marker expression studies were undertaken in both in vitro 4T1 and MDA MB 231 cell lines. The in vitro expression of Caspase 9 was upregulated while STAT3, VEGF, Cyclin D were downregulated when compared with GAPDH in cell lines treated with Cur and Pacli 10.0 μg/mL and in the combination of Au-CP at 5.0 μg/mL for 36 h. *p*-value: < 0.01 ** signifies very siginificant.

**Figure 5 ijms-23-02150-f005:**
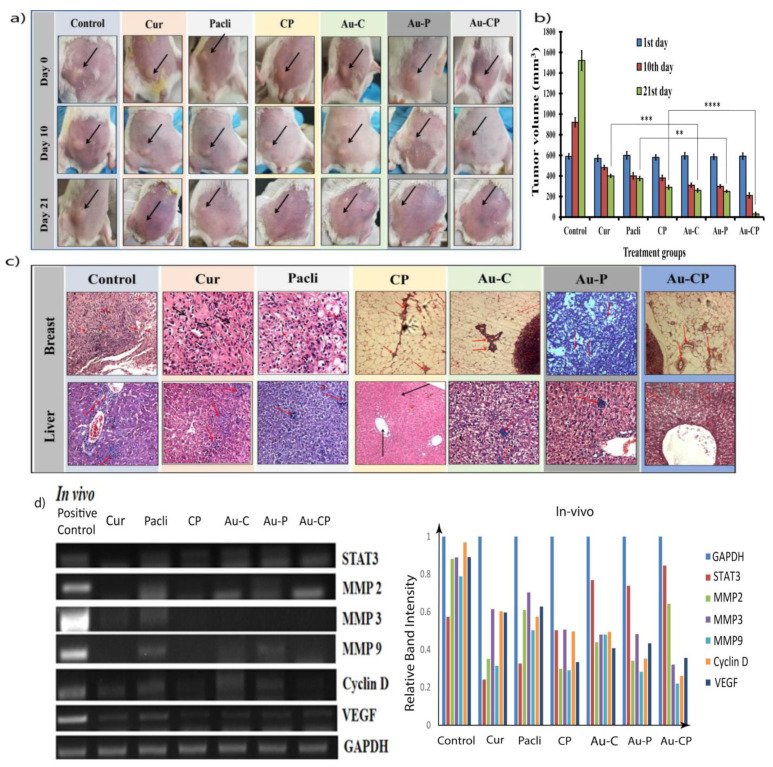
(**a**) Reduction in tumor size (primary xenografts) in BALB/c mice. BALB/c mice bearing 4T1 cells as xenografts were intraperitoneally treated with 50 mg/kg Cur and Pacli daily, individually and in combination with AuNP for 3 weeks. CP combination decreased tumor size more efficiently when conjugated with AuNP. (**b**) The size of tumors was measured externally in the experimental mice using a Vernier caliper at different time points (days 1, 10, and 21) and it was observed that the tumors regressed with the experimental treatments. (**c**) Histopathological evaluation of metastasis in breast fat pad and liver of the Balb/c mice that were treated with 50 mg/kg.bw Cur, Pacli, CP, Au-C, Au-P, and Au-CP. (**d**) The in vivo expression of STAT3, MMP2/3/9, Cyclin D, and VEGF was downregulated in rat tissues treated with 50 mg/kg Cur and Pacli individually and 25 mg/kg in combinations of CP with and without AuNPs groups [19]. **** *p*-value: <0.0001, *** *p*-value: <0.001, ** *p*-value: ≤0.01.

**Table 1 ijms-23-02150-t001:** Characterization of Au-C, Au-P, and Au-CP in water and cell culture media with serum.

Nanoparticles	In Water			In 10% Serum-Containing Medium		
	HDD (mm)	Zeta Potentials (mV)	PDI%	HDD (mm)	Zeta Potentials (mV)	PDI%
Au-C	101.5 ± 15	−0.2 ± 0.1	25.2 ± 2.5	152 ± 8	−6.9 ± 1.5	23.1 ± 1.5
Au-P	115 ± 9	−5.8 ± 2.1	23.9 ± 3.3	140 ± 15	−8.2 ± 2.3	20.2 ± 4.3
Au-CP	128 ± 10	−3.0 ± 1.1	25.5 ± 1.2	166 ± 6	−3.9 ± 1.12	22.41

## Data Availability

Data are contained within the article.

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
