# Peer review of "Modulatory Effects of Biosynthesized Gold Nanoparticles Conjugated with Curcumin and Paclitaxel on Tumorigenesis and Metastatic Pathways—In Vitro and In Vivo Studies"

_ijms, 2022, doi:10.3390/ijms23042150_

Round 1

Reviewer 1 Report

The authors have demonstrated a very important and relevant work, which, in addition to the fundamental, also has a significant applied character. However, the article has a number of significant shortcomings:

  1. The "Discussion" section is written very briefly. Basically, it is represented by a statement of the results obtained, and there is practically no explanation as to why Au-Cur-Pacli nanoparticles have the most effective anticancer activity in comparison with other combinations.
  2. In the "Results" section, the captions are very poorly readable in almost all the presented figures, it is necessary to increase the quality of the figures.
  3. In Figure 4 a-d it is necessary to do the statistical processing of the obtained results. It is not very clear what the authors wanted to demonstrate in Fig. 4 c-d: is this the result of DNA-electrophoresis of PCR fragments? Why is there no statistical processing?
  4. The authors should also discuss the mechanisms of action of other nanoparticles. https://pubmed.ncbi.nlm.nih.gov/34360564/ https://pubmed.ncbi.nlm.nih.gov/34439975/ 

Author Response

Thank you for submitting your manuscript for publication in International Journal of Molecular Sciences. The reviewer comments for the above-referenced manuscript are enclosed for your information. The reviewers indicate that the manuscript requires major revision to address a number of specific points before it can be published. On the basis of the reviewer comments and my own assessment of the manuscript, I am willing to consider a revised version of this paper for publication in International Journal of Molecular Sciences pending a second round of external review. In preparing the revision, carefully consider all of the comments made by the reviewers. We would like to receive your revision as soon as possible.

Reviewer 2 Report

Manuscript demonstrates interesting concept. However, it is not appropriately organized, poorly written at places and includes low quality figures. Please consider to address following comments:

  1. In abstract: Line 27, authors mention 'Second most reported cancer in both men and women..." The sentence is confusing and someone might miss interpret it as the breast cancer is second most reported cancer in men., which is not true. However, authors seems to imply in overall term (without classifying with gender). Please restructure sentence as it is confusing.
  2. In abstract line 40, author conclude "results suggest that the delivery of gold con-40 jugated Au-Curcumin-Paclitaxel formulations may...."If without gold particle, the combination is good enough, why do authors stress on gold conjugated curcumin-paclitaxel combination? Adding gold nanoparticles to Cur-Paclitaxel combination complicates the drug. 
  3. Introduction line 48-51: Confusing long sentence. 
  4. Introduction line 64: Inconsistency in manuscript, some places use E-cadherin, other places using E-cad.
  5. Manuscript is sloppy. Authors do not use a common abbreviation for treatments throughout manuscript- even within a paragraph it seems to change. 

    For example  on Page 4, last para. Authors use Cur, Pacli, Au-C, Au-P, CP, and Au-CP, AuNPs-CP at the beginning of the paragraph but use AuPacli, AuCur-Pacli etc. in line 156-157 of the same paragraph on Page 4. 

    Similarly, Figure 1c uses AuC, AuP, CP, and AuCP, but Figure 2 for example uses Au-C, Au-P, CP, and Au-CP. 

    Please make sure to use common abbreviations throughout the manuscript.  

  6. Table 1, it is unclear why larger diameter in 10% serum media. Authors did not explain. Authors should describe the reason and cite appropriate literature. 
  7. Axis on all figures are not clear. Please increase the figure resolution as well as font size. This is applicable to all figures in the manuscript as it is very hard to read data.
  8. MTT assay data: 

    Please normalize the OD values from control OD value to show relative % viability as usually shown for MTT assay in literature. 

    Also it is unclear in figure 1C what is 10 and 20? Author no where discuss results of MTT assay in the results section. 

  9. Line 153: Here authors just quantify results from Annexin-V/PI apotosis assay to classify cells into live and dead. Usually  Annexin-V/PI flowcytometry provides information to quantify cells in 4 different phases: live, early apoptosis, late apoptosis, and necrosis. I suggest authors to possibly represent their data in 4 different phases: live, early apoptosis, late apoptosis, and necrosis. This would add value to the manuscript. 
  10. Line 159: Authors write "...but the combination treatments demonstrated 159 better apoptotic efficiencies in comparison to the individual treatments....." Please conduct statistical tests and demonstrate if the combination caused significantly higher apoptosis than individual treatments using P-values. Such statements without proper statistical testing are often misleading. 

    Authors have conducted statistical comparison vs. control, may I suggest using ANOVA followed by multiple comparisons such as Tukey's test to demonstrate statistical difference among treatments throughout the manuscript? 

  11. Line 170: For flu intensity fold change calculation in DCFDA assay, was the quantification done by taking multiple images at different areas for a sample? If so, how many images per sample were taken? 

    - this question is significant because the cell distribution in slides could be different at different location of the slides between samples and if one random image was captured, the data here might not represent the actual outcome. 

    Moreover, were the # of cells similar between treatments after the cells were resuspended in 1x PBS. Was a counting done when cells were resuspended as described in methods. I would assume if the # of cells are different between samples, the flu will be different. 

    Please clarify and add this information into methods. 

  12. Typo in Figure 2a legend. 
  13. Manuscript not well organized. First author describe retardation in MDA-MB-231 cells migration in section 2.4 and just mention scratch assay without showing data and go to transwell migration then jump to E-cadherine expression in 2.5 but then come back again to scratch assay data on 2.6. It interrupts flow of manuscript. I suggest re-structuring of these results to make a good flow. 

  14. Similarly for in vivo data, authors first present gene expression data than tumor data. I suggest including in vivo gene expression data after tumor data in a separate figure. 
  15. Line 204, authors write "Whereas the Paclitaxel and Au-P treatments did not show any significant expres-204 sion signal suggesting its least efficacy against metastasis (Figure 3 C and D)" - Seems to be a contradictory statement. In figure 3c, authors shows significance as indicated by star. 

    Also it seems like there is increase in E-cad expression for these treatments and the difference appears to be large. 

    Please clarify. 

  16.  

    Quantification of E-cadherin shown for MDA-MB-231 but not for 4T1. Why?

  17. In line 217, it is unclear why author state "wound area increased..." I would assume that for the scratch assay the cells will try to fill the wound and area can remain same between 0h and 24h in best case.
  18. Is there a way authors can quantify the wound area in this assay? I think it would increase the quality of the manuscript. 

    For example: I came across a method using ImageJ for such quantification.

  19. Line 229 "We observed that the combinational treatments 229 significantly down-regulated VEGF, CYCLIN D1, and STAT-3 genes and up-regulated the 230 Caspase-9 expression in comparison to the individual treatments in both 4T1 and 231 MDAMB 231 cell lines." - 

    Authors show no quantification for these gene markers but include sentence "We observed that the combinational treatments significantly down-regulated VEGF, CYCLIN D1, and STAT-3 genes and up-regulated the 230 Caspase-9 expression in comparison..." 

    I recommend to include quantification of the bands. 

  20. Include # of mice across treatments in Figure 5a legend. 

    Moreover, please provide more information on animal model and tumor induction in the methods section, at least in brief so that article is self sufficient. 

  21. Line 421 in methods, authors use water fraction of curcumin. Curcumin is water insoluble, please clarify how would the water fraction of curcumin be helpful to make nanoparticles? 
  22. Line 451 in methods: 

    How the combination of Cur and Paclitaxel was made to be 10uM - does it mean that both Cur and Paclitaxel were 10uM each? Also how did author calculate the molarity of synthesized nanoparticles to create these stock solutions with specific molarity?- Please clarify. Easier would be to represent concentration of nanoparticle solutions in terms of ug/mL or mg/mL. 

    Authors need to provide more details on how these combinational stock solutions were made. 

    Also clarify whether it was 10uM or 10uM/ml? Both are different units and authors seem to use both interchangeably throughout the manuscript- which is unacceptable since 1 uM/mL = 1000 uM.

Author Response

(The authors gave the same response as above.)

Round 2

Reviewer 1 Report

I am very pleased with the latest version of the manuscript. I recommend accepting it for publication. I wish the authors success in their research.

Author Response

To

The Editor

International Journal of Molecular Sciences (IJMS)

We wish to extend our sincere thanks to the IJMS editorial team for giving us the opportunity to publish our research work in your reputed journal. As per the reviewer comments we made the necessary changes and resubmitting our manuscript for consideration in your renowned journal.

Reviewer(s)’ Comments to Author:

Reviewer #1: I am very pleased with the latest version of the manuscript. I recommend accepting it for publication. I wish the authors success in their research.

Response: we thank the reveiwer for reviewing our manuscript and giving his valuable suggestion cum recommendation to the journal to accept for our manuscript.

Reviewer 2 Report

Thanks for addressing my comments. 

  1. A minor typographic error in line 45. 'According' does not fit there as the sentence was started with it. Appears like a type-paste error.

2. Figures resolutions seems to be increased but it is still difficult to read the figure axis labels without squinting in some cases. I suggest authors to improve them further, if possible. 

Author Response

To

The Editor

International Journal of Molecular Sciences (IJMS)

We wish to extend our sincere thanks to the IJMS editorial team for giving us the opportunity to publish our research work in your reputed journal. As per the reviewer comments we made the necessary changes and resubmitting our manuscript for consideration in your renowned journal.

Reviewer(s)’ Comments to Author:

Reviewer #2:

  1. A minor typographic error in line 45. 'According' does not fit there as the sentence was started with it. Appears like a type-paste error.

Response: We thank the reviewer for his/her suggestion and for helping us in improving the manuscript. As per the given suggestion we have rectified our mistake and made necessary changes.

  1. Figures resolutions seems to be increased but it is still difficult to read the figure axis labels without squinting in some cases. I suggest authors to improve them further, if possible. 

Response: We apologize for inconvenience caused to the reviewer and also thank him/her for given suggestion, as per the suggestions we have further improved the image quality and also attaching the images in PDF for better resolution.
